# The Potential Role of Serotonergic Hallucinogens in Depression Treatment

**DOI:** 10.3390/life11080765

**Published:** 2021-07-29

**Authors:** Dominika Psiuk, Emilia Nowak, Krystian Cholewa, Urszula Łopuszańska, Marzena Samardakiewicz

**Affiliations:** Chair and Department of Psychology, Medical University of Lublin, 20-093 Lublin, Poland; emilia.m.nowak@wp.pl (E.N.); krystiancholewa1@wp.pl (K.C.); urszulalopuszanska@umlub.pl (U.Ł.); marzenasamardakiewicz@umlub.pl (M.S.)

**Keywords:** psychedelics, depression, serotonergic hallucinogens, psilocybin, MDMA, LSD, DMT

## Abstract

Due to an increasing number of depression diagnoses and limited effective treatments, researchers continue to explore novel therapeutic strategies for this disorder. Recently, interest has revolved around the use of serotonergic psychedelics to reduce the symptoms of depression. In this systematic review, we summarize the currently available knowledge on the safety and efficacy of psychedelic substances for the treatment of depression. A literature search of the PubMed/MEDLINE database identified 14 clinical trials from the last 10 years that examined the use of psilocybin, MDMA, DMT, or LSD for the treatment of depression symptoms. Some psychedelics, especially psilocybin, demonstrated an ability to reduce depressive symptoms as measured by several psychological scales, which was often sustained for months after the last psychedelic session. Moreover, one study revealed that psilocybin has comparable efficacy to escitalopram in the treatment of depression. None of the studies reported any serious adverse events associated with psychedelic administration. The reviewed studies suggest that psychedelics have great potential in depression therapy and, after addressing and overcoming the current study limitations, may be used as a novel method of treating depression in the future.

## 1. Introduction

### 1.1. Depression in the General Population

Depressive disorder is a demanding and common condition affecting more than 264 million people worldwide [1]. Despite many studies, the etiology of this disease remains unknown [2]. However, some factors are strongly associated with a higher risk of developing depression, including environmental factors like emotional, physical, and sexual abuse; genetic and epigenetic factors; and organic changes in the central nervous system, especially in the hippocampus. These factors can affect the neurobiological stress-responsive systems, resulting in neuroinflammation and altered neurotransmission [3]. The biological dysfunctions produced by these factors can also have a great impact on physical health and are associated with a higher risk of developing various conditions, such as heart disease, disability, diabetes mellitus, obesity, and cancer [4]. However, the most pressing clinical concern is suicide, a direct cause of death in patients with depressive disorders, as the pooled lifetime prevalence of suicide attempts in depressed patients is estimated to be 27–34%, which is almost 20 times greater than in general population [1,5].

### 1.2. Current Treatment Methods for Depression and Their Limitations

Currently, the two main treatment methods for depression are psychotherapy and pharmacotherapy. There are various types of psychotherapies with proven efficacy, including cognitive-behavioral therapy, behavioral activation therapy, problem-solving therapy, and interpersonal therapy [6]. Pharmacotherapy is the first-line therapy, and it is based on antidepressants, such as monoaminergic reuptake inhibitors and tricyclic antidepressants [7]. Despite proven efficacy, a high percentage of patients treated with these drugs exhibit treatment-resistant depression [8]. Indeed, it is estimated that only about 30–40% of patients with depression, especially those diagnosed with major depressive disorders (MDD), acquire full remission with first-line therapy, and about one third of patients do not achieve remission even if treated with as many as four different antidepressants [9]. Antidepressant treatment is also associated with a wide array of adverse events [10]. Although current research suggests that fluoxetine, a serotonin reuptake inhibitor, is likely the only effective antidepressant for children and adolescents, the WHO discourages antidepressant use in children and recommends against its use as a first-line treatment in adolescents [11,12]. Often both psychotherapy and pharmacotherapy are combined. The main advantages of non-pharmacological treatments are their safety profile and low incidence of adverse effects. However, the quality of the evidence indicates their efficacy is poor [13]. There are other methods used for the treatment of depression, such as electroconvulsive therapy, brain stimulation, physical activity, and even biological treatment [14,15,16,17,18]. Among brain stimulation techniques worth mentioning are cranial electrotherapy stimulation (CES) and repetitive transcranial magnetic stimulation (rTMS), both FDA-approved for major depressive disorder [15,19,20,21]. Currently, many researchers are examining the potential of using serotonergic hallucinogens to reduce depressive symptoms.

### 1.3. Serotonergic Hallucinogens as a Potential Treatment for Depression

Hallucinogens are a group of psychoactive chemicals with various mechanisms of action that cause significant alterations in the central nervous system, resulting in changes in human consciousness [22]. These substances, such as d-lysergic acid diethylamide (LSD), dimethyltryptamine (DMT), and psilocybin, can act on serotonin 5-HT2A receptors as agonists. More recently, chemists have developed new “designer drugs”, such as 3,4-methylenedioxymethamphetamine (MDMA), that stimulate the release of serotonin, dopamine, and norepinephrine and can inhibit their reuptake by blocking, for example, the serotonin transporter [23]. Hallucinogens may also have additional actions at other serotonin receptors (particularly 5-HT1A and 5-HT2C) and can act on other neurotransmitter systems [24,25]. These drugs can also exhibit rapid and long-lasting antidepressant and anxiolytic effects [26]. It also seems as if the experience of an altered state of consciousness produced by these agents has a great impact on the final therapeutic effect [27]. It should also be noted that hallucinogens do not lead to dependence, as these drugs do not directly influence the dopaminergic system [28].

Over the past decade, numerous studies have examined the use of psychedelics to treat various conditions, many of which have suggested that these substances might be promising agents for treating depression, as an addition to pharmacotherapy and combined with psychotherapy, the two first-line treatments. To aid in future investigations and the potential approval of these substances for clinical use, we examine here the evidence from recent clinical trials on the safety and efficacy of psychedelics for the treatment of depressive symptoms.

## 2. Materials and Methods

The aim of this work is to review the potential use of the hallucinogens psilocybin, MDMA, DMT, and LSD in the treatment of depression. Based on the guidelines provided by the Primary Reporting Items for Systematic Reviews and Meta-Analyses Statement, the PubMed/MEDLINE database was used to identify potential articles for analysis using the following search terms: (depression) AND (lysergic diethylamide) OR (psilocybin) OR (MDMA) OR (dimethyltryptamine)). The literature search was conducted on 13 May 2021, and 817 items were obtained.

The results were filtered for time (last ten years, 321 results), character of the studies (clinical trials, 23 results), and participant type (human participants only, 23 results). The following types of studies were excluded: questionnaire-based surveys, surveys specifying the state of the respondents’ knowledge, meta-analyses, and reviews. Only studies that examined LSD, psilocybin, MDMA, or DMT, were included as the main focus of this review was to examine the influence of these substances on depressive symptoms.

Next, two people read the abstracts of the identified studies and excluded those that did not meet the selection criteria. As a result of the above searches, a total of 14 studies were included for review. A total of 8 of these studies were randomized controlled trials, and 6 were open-label studies. The selection process is illustrated in Figure 1.

## 3. Review

### 3.1. Psilocybin

There were four studies [29,30,31,32] that evaluated the efficacy and safety of using psilocybin in the treatment of depression symptoms. The research protocol of each study was similar and consisted of two 6-hour experimental sessions spaced a few (2–7) weeks apart. The sessions were conducted in a room designed to be aesthetic and appealing to the participants. During the sessions, the participants were encouraged to lie in bed and listen to preselected music. They were also monitored by health professionals, who supported and encouraged the participants to relax, trust, and follow the experience and helped them to sum up their thoughts and feelings after the sessions. After the sessions, the patients discussed their subjective cognitive, affective, and psychospiritual experiences. All four studies were cross-over designs, and patients acted as their own controls. The characteristics of each study are presented in Table 1.

The study of Kraehenmann et al. focused on the effects of psilocybin on amygdala reactivity to negative stimuli obtained from the International Affective Picture System (IAPS). The study involved 25 healthy volunteers who, in large part, had no history of previous hallucinogen use. The participants received either psilocybin (0.16 mg/kg) or placebo (lactose) in two experimental treatment sessions. During the sessions, the mood was assessed, and functional magnetic resonance imaging (fMRI) was used to evaluate the effects of psilocybin on amygdala activity during emotion processing. The results showed that psilocybin reduced amygdala reactivity to negative and neutral stimuli and improved the mood of healthy volunteers, as confirmed by the Positive and Negative Affect Schedule (PANAS) scores. These findings suggest that psilocybin may normalize the amygdala hyperactivity and negative mood states associated with depression [29].

In the Ross et al. study, 29 patients with a diagnosis of advanced cancer received either psilocybin (0.3 mg/kg) or active control (niacin, 250 mg) in two experimental treatment sessions. Before the cross-over, there were significant differences in anxiety and depression between the psilocybin and control groups, as measured by several scales (Hospital Anxiety and Depression Scale [HADS], Beck Depression Inventory [BDI], and State-Trait Anxiety Inventory [STAI]). The reductions in anxiety and depression remained significant at each time point, including the final point at 8 months after the second dose. Seven weeks after the first dose, 83% of patients in the psilocybin group met the criteria for an antidepressant response as measured by the BDI, compared with 14% in the control group. In addition, 58% of the participants administered psilocybin exhibited an anxiolytic response compared to 14% of the controls. At a 6.5-month follow-up, the antidepressant or anxiolytic response rates were approximately 60–80%. These results show that psilocybin produces immediate and enduring antidepressant and anxiolytic responses. In addition to these effects, 2 weeks after the first dose, psilocybin decreased cancer-related demoralization and hopelessness and improved quality of life and spiritual wellbeing, effects that were sustained at the final follow-up. It was also shown that the psilocybin-induced mystical experience, measured with the Mystical Experience Questionnaire (MEQ 30), had an impact on the therapeutic effects of psilocybin on anxiety and depression [30].

In the Griffiths et al. study, 51 patients with cancer and depression and anxiety symptoms received both low-dose, placebo-like doses of psilocybin (1 or 3 mg/70 kg) and high doses of psilocybin (22 or 30 mg/70 kg) in the first or second session. Psilocybin significantly reduced the depression and anxiety symptoms, as assessed by both clinical-rated and self-rated scales. The overall rates of clinical response at a 6-month follow-up were 78% and 83%, while the overall rates of symptom remission were 65% and 57% for depression and anxiety, respectively. Moreover, the participants reported that the experience positively influenced their life by changing their attitudes about themselves, relationships, and spirituality. No serious adverse effects were noted [31].

The most recent study, conducted by Carhart-Harris et al., was the first study that aimed to compare the antidepressive effects of psilocybin and a drug already approved for depression treatment, escitalopram. Sixty-six patients that met the DSM-IV criteria for MDD were randomized and allocated to two groups. One group received a placebo for 6 weeks and the other 10 mg and 20 mg of escitalopram for the first 3 weeks and the last 3 weeks, respectively. 25 mg of psilocybin was given to the placebo group in two separate sessions, one before the treatment and the second after 3 weeks of treatment, while the escitalopram group received 1 mg psilocybin in each session. The change in the Quick Inventory of Depressive Symptomatology-Self-Report (QIDS-SR-16) score at 6 weeks was the primary endpoint and was −8 versus −6 in the psilocybin and escitalopram groups, respectively, but this difference did not reach statistical significance. Using the same scale, the response and remission rates measured at 6 weeks were 70% versus 48%, and 57% versus 28% for the psilocybin and escitalopram groups, respectively. Other secondary measures of depression were also in favor of psilocybin. However, the confidence intervals were not corrected for multiple comparisons. No serious adverse effects were reported in either group [32].

### 3.2. MDMA

Multiple studies have examined the effectiveness of MDMA for treating post-traumatic stress disorder (PTSD) [37,38,39,40], but few have focused on depression. Among the clinical trials concerning MDMA-assisted psychotherapy that are available on the PubMed database, only three examined the influence of MDMA administration on depression symptoms.

The Ot’alora et al. study involved 28 patients with chronic PTSD, of which nearly half (42.9%) had been diagnosed with MDD and 25% with depression. The lifetime Suicide Severity Rating Scale (SSRS) showed that 27/28 (92.6%) patients had suicidal ideation, and 8/28 (28.6%) exhibited suicidal behavior. During two eight-hour sessions spaced a month apart, patients received either an active (125 or 100 mg) or a comparator (40 mg) dose of MDMA. Changes in depressive symptoms, as measured by the BDI, showed approximately equivalent decreases across the groups. However, after an additional third open-label session where all participants received 100 mg or 125 mg MDMA, there was a significant improvement in the BDI at the 12-month follow-up compared to the baseline (7.3 points vs. 27.8 points) [33].

In 2019, Mithoefer et al. conducted an analysis of six phase 2 clinical trials. The study showed a significant reduction in symptoms in a large sample of PTSD patients (n = 103) treated with active doses of MDMA combined with psychotherapy. Patients from the active groups received 75–125 mg of MDMA, and those in the control groups received doses of 0–40 mg. The change in depression symptoms, as measured by the BDI scale, was −12.4 versus −6.5 for the active and control groups, respectively, and the difference between groups was statistically significant (*p* = 0.053) [34].

Most recently, Wolfson et al. conducted a study examining MDMA-assisted psychotherapy for psychological distress related to life-threatening disease treatment [38]. In this trial, 14 out of 18 patients were diagnosed with MDD. After two blinded experimental sessions, the mean changes in the BDI-II and Montgomery–Asberg Depression Rating Scale (MADRS) scores were −14.6 versus −20.9, and −7.0 and −10.5, for the placebo and MDMA groups, respectively, but the differences did not reach statistical significance. However, the results obtained after a third open-label session where both groups received MDMA were statistically significant. The total outcome scores for the BDI-II and MADRS after treatment, and at a 6- and 12-month follow up, were 3.0, 3.2, 4.3 (from the baseline 30.2) and 4.1, 3.6, 3.5 (from the baseline 19.4), respectively. The changes were similar in both groups but more strongly expressed in the MDMA group. The MDMA treatment was well tolerated, and no serious adverse events were reported [35].

### 3.3. LSD

Bershad et al. conducted a double-blind study where 20 healthy volunteers received 0, 6.5, 13, or 26 μg of LSD, each in one out of four 8-hour sessions. Subjective mental health state was measured at the baseline, at 30 to 90 min intervals after receiving the drug, and at the peak of the drug effect. The sessions were spaced at least 7 days apart. The Profile of Mood States (POMS) was used to measure anxiety and depression symptoms. The POMS anxiety scale revealed that the LSD effect correlated with the dose administered and that there was a trend for the highest dose to increase ratings. However, for the POMS depression scale, the effects of LSD did not reach statistical significance [36].

### 3.4. Open-Label Studies

In this review, we also included six open-label studies, as detailed in Table 2 [41,42,43,44,45,46]. Five studies revealed that the administration of 10–25 mg psilocybin in patients with treatment-resistant MDD significantly reduced depressive symptoms and that this effect persisted for 5 weeks–5 months as measured by the QIDS, BDI, STAI-T and Snaith-Hamilton Pleasure Scale (SHAPS) [41,42]. fMRI images also showed that the amygdala response to emotional stimuli increased after psilocybin intake, which can be related to an enhanced ability to face and work through negative emotions, an increase in functional connectivity between the amygdala and prefrontal cortex and the occipital-parietal cortices, and a revival of emotional responsiveness [44,45].

One open-label study also examined how DMT influences depression symptoms, although this was not the primary endpoint. This study aimed to assess the effects of inhaled DMT on neuroendocrine markers, affect, mindfulness, and stress biomarkers. Eleven participants were enrolled on the study, and 10 of them also completed the online follow-up assessment. Each participant had an individual session where one to four DMT doses were inhaled (17 mg to 61 mg, respectively). Participants completed five questionnaires measuring mental health, with the most relevant to this review being the Depression, Anxiety, Stress Scale 21 (DASS-21). There was a significant reduction in depression symptoms immediately post-session, which was sustained and approached significance at a 7-day follow-up. No serious adverse reactions were noted. However, three patients reported feeling fear, confusion, anger, or shock, one reported scratching in their throat, and one patient vomited shortly after DMT intake [46].

### 3.5. Adverse Events after Hallucinogen Use in the Reviewed Studies

Adverse events from the reviewed studies were summarized and are shown in Table 3. None of the studies revealed any serious adverse events that occurred during the psychedelic administration period or afterwards. In the table, we included only studies that reported a profile of adverse events. Five out of fourteen studies did not include any information about adverse effects [29,36,43,44,45]. The reported adverse events were divided into five categories: psychological, neurological, cardiovascular, gastroenterological, and general. The most common psychological adverse events were transient anxiety [30,33,34,41,42,46] and psychological discomfort [31], while the most common neurological events were headache [30,32,33,34,41,42] and, for MDMA administration, dizziness and jaw clenching [33,34]. In the psilocybin studies, there were also gastroenterological adverse events noted, most commonly nausea and/or vomiting [30,31,32,41,42]. Most researchers measured blood pressure and heart rate during psilocybin studies and observed an elevation in blood pressure [30,31] and heart rate [30]; however, these increases were often not considered adverse events [32,33,34,35,36,41,42]. There were also some general adverse events noted, mostly in MDMA studies, including fatigue [32,33,34] and a lack of appetite [33,34,35].

## 4. Discussion

The efficacy of serotonergic hallucinogens depends on two properties: the molecular or pharmacological aspect, which is an agonist action at the 5-HT2A receptor; and the strongly related psychological aspect, namely the ability to produce highly intense experiences of altered consciousness [47]. Serotonin 5-HT2A receptors are known for their roles in brain plasticity, cognition, mood, and perception [48]. Due to agonist activity at these receptors, psychedelics cause multifaceted actions, such as reducing amygdala reactivity, reducing the levels of pro-inflammatory cytokines, and reducing threat sensitivity in the visual cortex, which are responsible for the antidepressant and anxiolytic outcomes [49]. Psychedelics are also known to reduce connectivity in default mode networks by triggering a temporary disintegration of canonical “resting-state” networks—in other words, these agents increase global functional integration [23]. This effect results in the experience of various hallucinations and so-called “ocean boundlessness”, which comprises a self-dissolution state and deep feelings of unity and self-transcendence, a sense of intuitive understanding, and heightened mood. As the experience continues, sensations become more personalized, which often cause the recall of past memories, and eventually, strong emotions start to appear [50]. Considering the above, it is believed that the psychedelic experience, next to the substances’ molecular activity, has a great impact on therapeutic outcomes [27].

In most studies, psychedelic administration led to a reduction in depressive symptoms [33,34,35,36,42,43,44]. Among the randomized controlled trials, all four studies with psilocybin, three with MDMA, and one with DMT demonstrated significant efficacy in reducing the symptoms of depression and in reducing amygdala hyperactivity, a distinctive feature of depression. In the study with LSD, a reduction in depressive symptoms was also observed among participants who received the psychedelic substance; however, the effects did not reach statistical significance [36]. Authors of meta-analyses concerning topics similar to ours have also observed positive outcomes of using psychedelics in reducing depressive symptoms [51,52]. Galvão-Coelho et al. distinguished acute (between 3 h and 1 day after dosage), medium (2–15 days) and long-term (16–60 days) effects for reduction of depressive symptoms and mood state. They observed moderate acute effect size and large long-term effects of psilocybin in reducing depressive symptoms and moderate acute effects after psilocybin and LSD administration in reducing negative mood compared to placebo, both in healthy volunteers and patients with mood disorders. In comparison to psilocybin, LSD demonstrated a larger reduction of negative mood [51]. Romeo et al. calculated the mean percentages of improvement in depression score from baseline after psychedelic (psilocybin, ayahuasca, and LSD) treatment. The decrease of depressive symptoms was the highest at day 21 (66.8%) and the lowest at weeks 6–8 (34.2%). On both day 1 and month 6, the decrease was 49.6%. Authors also observed more exacerbated subjective drug effects, such as more important mystical experiences, in patients who received higher doses of psychedelic substance and suggested the possible relationship between the dose and the predicted therapeutic outcome of administering psilocybin [52]. It is worth mentioning that both authors included ayahuasca, a source of DMT, in their studies. In our study, we decided not to, mainly because the ayahuasca solution contains other than DMT components, such as harmine, harmaline, and tetrahydroharmine. In addition, in studies with ayahuasca, the dose of DMT was often not stable [53,54,55]. On the other hand, neither of those two studies included MDMA, while we decided otherwise, considering the positive outcomes in reducing depressive symptoms among patients with PTSD [33,34].

Attention should be paid to the latest study conducted by Carhart-Harris, which was the first trial to compare the efficacy and safety of psilocybin with an approved antidepressant drug [32]. In this study, therapy with psilocybin demonstrated comparable efficacy to escitalopram, with higher response and remission levels than escitalopram [32]. These findings suggest that psilocybin will be considered as a therapy option for depression in the future. However, pharmacological treatment should remain the first-line therapy, as its efficacy and safety are proven in both clinical practice and in many conducted metanalyses in the past [56,57,58].

Moreover, if administered in properly prepared and safe conditions, serotonergic hallucinogens demonstrate only mild and temporary adverse events, such as transient anxiety, headache, dizziness, and nausea (Table 3), whereas antidepressants drugs are associated with a wide range of persistent adverse events, such as weight gain [59], sexual dysfunction [60], somnolence, and insomnia [61]. Antidepressant drugs can also significantly interact with other medications [62]. The number of side effects is related to depression severity and a longer duration of treatment [63], and, while the first effects of antidepressant therapy appear after 2 weeks of treatment, the recommended time to achieve remission is 0–6 months, and 6–24 months to prevent recurrence and a return to normal functioning [64]. Psychedelic-assisted therapy, by contrast, requires administering the drug only once every few weeks over two to three individual sessions, and the effects are immediate and sustained for at least 6–12 months [33,65].

It is important to mention that although none of the clinical trials included in this review revealed any serious adverse events occurred during the psychedelic administration period and afterwards, researchers detected some cases of serious adverse events caused by serotonergic substances, foremost MDMA, such as serotonin syndrome [66], hallucinogen persisting perception disorder (HPPD) [67], exogenous psychosis [68], fatal hyperthermia [69], rhabdomyolysis, and renal failure [70]. Moreover, MDMA is reported to cause suicidality [71]. Therefore, the possible risk of using psychedelics cannot be ruled out, and their administration requires particular caution.

There were a few limitations in the reviewed studies. First, the sample size was generally small in each study and ranged from 10 to 103, with an average of 28. In addition, 3 [29,41,46] out of the 14 studies included healthy volunteers as participants, which limits the generalizability of the results. Moreover, 6 out of the 14 studies [41,42,43,44,45,46] were open-label studies, which can affect the reliability of the results. However, due to the difficulties in constructing the control group in psychedelic studies, we decided to include their results in this review.

Due to the powerful effects and inimitable sensations accompanying the intake of psychedelic substances, it is virtually impossible to design a trial with a control group that can fulfill the double-blind standard for clinical studies. Consequently, studies examining the therapeutic potential of psychedelic substances face an important methodological problem. Researchers have attempted to overcome this limitation by administering an active placebo (e.g., niacin) [26] or low, placebo-like doses of the studied substance [31,32,33,34] to control groups, which is by far the most efficient method of maintaining blinding in these studies.

As mentioned above, the psychedelic-induced experience often involves very intense, psychosis-like effects, which can lead to an anxious ego-dissolution (the so-called “bad trip”) that is characterized by a feeling of losing control over one’s own body and autonomy, panic, and cognitive impairment [50]. As the effects of psychedelic substances are long-lasting (from 6 to 12 h, depending on the substance and the dose), to ensure patient safety, the duration of each session needs to last at least as long as the psychedelic effect is maintained [30,33,36,41,56]. Thus, it is understandable that, for many patients, the experience can be too overwhelming and the sessions too time-consuming to attend. In the aim to minimize the anxiety before psychedelic sessions, each must be thoroughly prepared, individually for each patient, in the company of an experienced monitor whom the patient knows and trusts. There are also some medical conditions that exclude participation in psychedelic sessions, including schizoid and paranoid traits or even a family history of schizophrenia [72].

It is also imperative to note the importance of set (expectation, preparation, intention) and setting (physical and social environments) as determinants of the psychedelic experience [73]. Properly prepared patient and session environments are crucial for the therapeutic outcome and can reduce the occurrence of “bad trips” [74]. Modifications of the set can include preparation for the session, monitoring during the session, assurances of interpersonal support, and discussion after the session to summarize the experience [25]. Therefore, the presence of a qualified monitor/psychotherapist is critical for a safe and effective therapeutic session. It should be noted that, despite the long-lasting outcomes, psychedelic sessions should be considered as a part of a longer therapeutic process rather than a single experience. As mentioned in the introduction, there is little abuse potential or risk of dependence, which is precisely due to the intense and long-lasting effects the psychedelics produce. Moreover, they cause an immediate tolerance; thus, taking one dose after another will result in very little effect [75]. However, to remove any risk of abuse, psychedelics should be mandatorily administered under supervision, only on the recommendation of a psychiatrist or psychotherapist qualified in this field and only in the company of a monitor in a safe environment. The setting of the sessions should take place in an aesthetically pleasing room with a comfortable place for resting, such as a sofa or bed. In some studies, participants wore eye masks and listened to preselected music. Participants should be encouraged to lie down, relax, trust the process, and focus their attention on their inner experiences. After each session, participants and monitors should also discuss the experience [31].

## 5. Conclusions

The studies reviewed above suggest that some hallucinogens are effective at reducing depressive symptoms and indicate that these agents may be used in the future as a novel treatment for depression. The administration of hallucinogens, especially psilocybin, results in a sustained reduction in depressive symptoms with an absence of serious adverse effects. It is worth mentioning the importance of the psychedelic experience itself and realizing that it can often be intense and overwhelming for a patient. Thus, the presence of a properly qualified monitor-therapist throughout the experience is necessary to ensure a positive result from each session. Compared to currently available methods, psychedelic therapy seems to be safer than the chronic use of antidepressant drugs but requires more time than regular therapy, as the effects of psychedelics can last up to 12 h. However, antidepressants and psychotherapy should remain first-line treatments, as their efficacy and safety are validated in clinical practice and available literature.

There are a few limitations of the existing studies on psychedelics. First, the sample size used was small and often consisted of healthy volunteers rather than patients suffering from actual conditions. Due to the characteristic and inimitable effects of these drugs, it has also been difficult to construct a study with adequate double blinding. To overcome this problem, some researchers have used a low, “placebo-like” dose of a substance for the control group, which can produce some effects that may not be obvious to the participants.

## Figures and Tables

**Figure 1 life-11-00765-f001:**
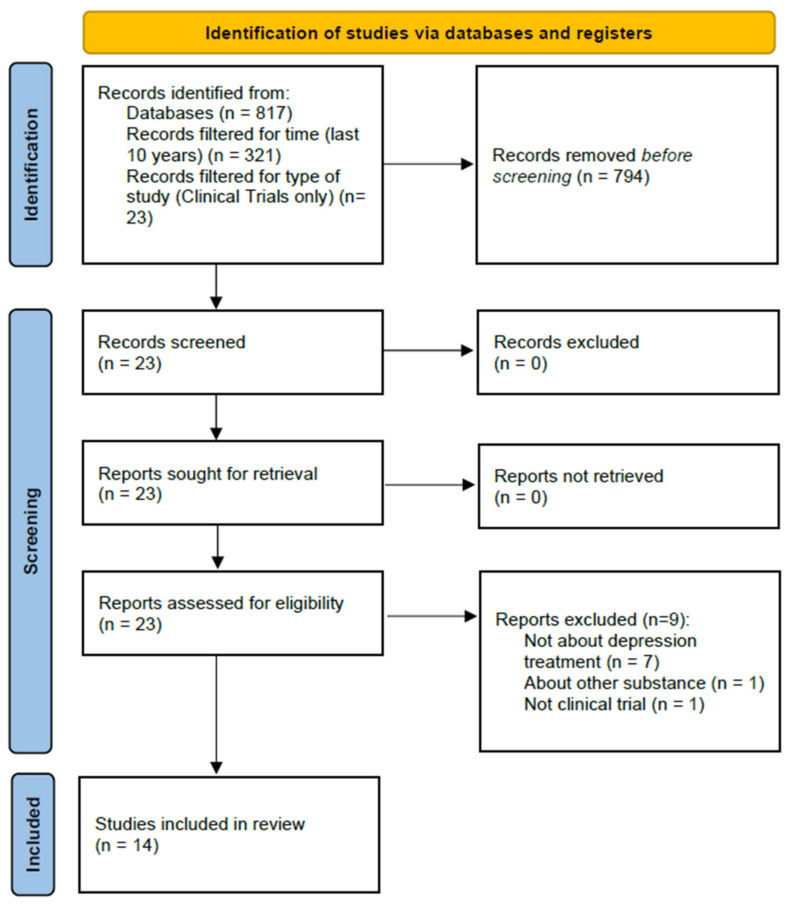
Article selection process.

**Table 1 life-11-00765-t001:** Characteristics of reviewed studies on the use of MDMA, DMT, and LSD in the treatment of depression (Randomized Controlled Trial study, RTC).

No.	Studies’ Authors, Year	Substance/Test Group	Substance/Control Group	Comorbid Condition	No. of Participants	Used Questionnaires to Measure Depressive Symptoms
1	Kraehenmann R et al., 2015 [29]	Psilocybin, 0.16 mg/kg	Lactose	Healthy volunteers	25	PANAS, STAI
2	Ross S et al., 2016 [30]	Psilocybin, 0.3 mg/kg	Niacin, 250 mg	Advanced cancer	29	HADS, BDI, STAI, MEQ 30
3	Griffiths RR et al., 2016 [31]	Psilocybin, 22 or 30 mg/70 kg	Psilocybin, 1 or 3 mg/70 kg	Advanced cancer	51	GRID-HAMD-17, BDI, HADS, STAI
4	Carhart-Harris R et al., 2021 [32]	Psilocybin, 25 mg versus Escitalopram, 10–25 mg	Psilocybin 1 mg, placebo	Major depressive disorder	66	QIDS-SR-16
5	Ot’alora G M et al., 2018 [33]	MDMA, 125 or 100 mg	MDMA, 40 mg	PTSD	28	BDI
6	Mithoefer MC et al., 2019 [34]	MDMA, 75–125 mg	MDMA, 0–40 mg	PTSD	103	BDI
7	Wolfson PE et al., 2020 [35]	MDMA, 125 mg	Lactose, 125 mg	Life-threatening illness	18	BDI, MADRS
8	Bershad AK et al., 2019 [36]	LSD, 6.5–26 μg	Lactose	Healthy volunteers	20	POMS

Abbreviations: 5D-ASC—5-Dimension Altered States of Consciousness; BDI—Beck Depression Inventory; BPRS—Brief Psychiatric Rating Scale; DASS-21—Depression, Anxiety, Stress Scale 21; DMT—dimethyltryptamine; GRID-HAMD-17—GRID-Hamilton Depression Rating Scale; HADS—Hospital Anxiety and Depression Scale; MADRS—Montgomery–Asberg Depression Rating Scale; MDMA—3,4-methylenedioxymethamphetamine; MEQ 30—Mystical Experience Questionnaire; LSD—D-lysergic diethylamide; PANAS—Positive and Negative Affect Schedule; POMS—Profile of Mood States; STAI—State-Trait Anxiety Inventory; QIDS-SR-16—Quick Inventory of Depressive Symptomatology-Self-Report.

**Table 2 life-11-00765-t002:** Characteristics of reviewed studies on the use of MDMA, DMT, and LSD in the treatment of depression (open-label studies).

No.	Studies’ Authors, Year	Substance	No. of Participants	Comorbid Condition	Results
1.	Carhart RL et al., 2016 [41]	Psilocybin, 1. dose: 10 mg, 2. dose: 25 mg	12	Major depression disorder (MDD) (moderate to severe degree, treatment-resistant)	A significant reduction in depressive symptoms lasting up to 3 months after 2. dose, relative to the baseline, measured by: QIDS (10.0 vs. 19.2) BDI (15.2 vs. 33.7) STAI-T (54.8 vs. 70.1) SHAPS (2.8 vs. 7.5)
2.	Carhart RL et al., 2017 [42]	Psilocybin, 1. dose: 10 mg, 2. dose: 25 mg	19	MDD (treatment-resistant)	A significant reduction in depressive symptoms lasting up to 5 weeks, measured by QIDS-SR16, were observed in 18 patients (95%) (mean change: −9.2); 9 patients (47%) met criteria for response (≤50% reductions). Reduced depressive symptoms were correlated with decreased amygdala CBF in fMRI.
3.	Carhart RL et al., 2018 [43]	Psilocybin, 1. dose: 10 mg, 2. dose: 25 mg	19	MDD (severe, treatment-resistant)	A significant reduction in depressive symptoms lasting up to 6 months, relative to the baseline, measured by: QIDS-SR16 (*p* = 0.0035) BDI (19.5 vs. 34.5) STAI (53.8 vs. 68.6)
4.	Roseman L et al., 2018 [44]	Psilocybin, 1. dose: 10 mg, 2. dose: 25 mg	20	MDD (moderate to severe, treatment-resistant)	Post-treatment increased amygdala responses to emotional stimuli in fMRI suggest that psilocybin allows to confront and work through negative emotions (induced by showing fearful and happy faces).
5.	Mertens LJ et al., 2020 [45]	Psilocybin, 1. dose: 10 mg, 2. dose: 25 mg	19	MDD (moderate to severe, treatment-resistant)	Post-treatment increase in functional connectivity between the amygdala and ventromedial prefrontal cortex to occipital-parietal cortices in fMRI during face processing.
6.	Uthaug MV et al., 2020 [46]	DMT, 17–61 mg	10	-(Healthy volunteers)	Significant reduction of depressive symptoms, sustained at 7-day follow-up, measured in DASS-21.

Abbreviations: BDI—Beck Depression Inventory; CBF—Cerebral Blood Flow; DASS-21—Depression, Anxiety, Stress Scale 21; fMRI—functional Magnetic Resonance Imaging; QIDS—Quick Inventory of Depressive Symptoms; QIDS-SR16—16-item Quick Inventory of Depressive Symptoms; SHAPS—Snaith-Hamilton Pleasure Scale; STAI-T—State-Trait Anxiety Inventory.

**Table 3 life-11-00765-t003:** Adverse events after use of hallucinogens in analyzed studies.

	Clinical Trial				Adverse Events (AEs)				
No.	Study	No. of Participants	Substance	Dose	Psychological	Neurological	Cardiovascular	Gastroenterological	General
1.	Ross S et al., 2016 [30]	29	psilocybin	0.3 mg/kg	transient anxiety: 17%, transient psychotic-like symptoms: 7%	headache/migraine: 28%	elevation in blood pressure (BP) and heart rate (HR): 76%	nausea: 14%	-
2.	Griffiths RR et al., 2016 [31]	51	psilocybin	22 or 30 mg/70 kg	psychological discomfort: 32%, anxiety: 26%, paranoid ideation: 2%	headache: 2%	elevation in systolic BP: 34%, elevation in diastolic BP: 12%	nausea/vomiting: 15%	physical discomfort: 21%
3.	Carhart RL et al., 2016 [41]	12	psilocybin	10 and 25 mg	transient anxiety: 100%, thought disorder: 75%	headache: 33%	-	nausea: 33%	-
4.	Carhart RL et al., 2018 [42]	19	psilocybin	10 and 25 mg	transient anxiety: 79%, transient paranoia: 16%	headache: 42%	-	nausea: 26%	-
5.	Carhart RL et al., 2021 [32]	30	psilocybin	25 mg	feeling jittery: 7%, sleep disorder: 3%	headache: 67%, migraine: 10%	palpitations: 3%	nausea: 27%, vomiting: 7%, diarrhea: 3%	fatigue: 7%
6.	Ot’alora GM et al., 2018 [33]	28	MDMA	125 mg	During experimental session anxiety: 54%	During experimental session dizziness: 54%, headache: 23%, jaw clenching: 62%, muscle tension: 54%	-	-	During experimental session fatigue: 31%
					During 7 days’ observation anxiety: 77%, depressed mood: 15%, insomnia: 46%	During 7 days’ observation headache: 39%, muscle tension: 46%		During 7 days’ observation nausea: 62%	During 7 days’ observation fatigue: 69%, lack of appetite: 62%
				100 mg	During experimental session anxiety: 67%	During experimental session dizziness: 22%, headache: 44%, jaw clenching: 56%, muscle tension: 44%	-	-	During experimental session fatigue: 44%
					During 7 days’ observation anxiety: 89%, depressed mood: 22%, insomnia: 78%	During 7 days’ observation headache: 33%, muscle tension: 11%		During 7 days’ observation nausea: 33%	During 7 days’ observation fatigue: 78%, lack of appetite: 11%
				40 mg	During experimental session anxiety: 33%	During experimental session dizziness: 17%, headache: 67%, jaw clenching: 33%, muscle tension: 33%	-	-	During experimental session fatigue: 33%
					During 7 days’ observation anxiety: 33%, depressed mood: 0%, insomnia: 50%	During 7 days’ observation headache: 67%, muscle tension: 33%		During 7 days’ observation nausea: 17%	During 7 days’ observation fatigue: 33%, lack of appetite: 17%
7.	Mithoefer MC et al., 2019 [34]	103	MDMA	75–125 mg	anxiety: 72%	dizziness: 40%, headache: 53%, jaw clenching: 64%	-	nausea: 40%	fatigue: 49%, lack of appetite: 49%
				0–40 mg	anxiety: 48%	dizziness: 19%, headache: 71%, jaw clenching: 19%	-	nausea: 19%	fatigue: 58%, lack of appetite: 23%
8.	Wolfson PE et al., 2020 [35]	18	MDMA	125 mg	During experimental session anxiety: 23%, insomnia: 15%	During experimental session headache: 62%, jaw clenching: 85%	-	During experimental session nausea: 23%	During experimental session dry mouth: 69%, perspiration: 69%, thirst: 85%, lack of appetite: 31%
					During 7 days’ observation anxiety: 62%, insomnia: 69%, low mood: 62%	During 7 days’ observation jaw clenching: 62%		During 7 days’ observation nausea: 46%	During 7 days’ observation dry mouth: 23%, fatigue: 92%, lack of appetite: 31%
9.	Uthaug MV et al., 2020 [46]	10	DMT	17–61 mg	anxiety: 20%, insomnia: 10%	muscle tension: 20%	-	vomiting: 10%	physical distress: 10%

## Data Availability

Not applicable.

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
