# Peer review of "The Potential Role of Serotonergic Hallucinogens in Depression Treatment"

_life, 2021, doi:10.3390/life11080765_

Round 1
Reviewer 1 Report
Basically, this is an interesting paper for the effects of hallucinogens for the treatment of depression. However, there are several points to be improved:
1. As a systemic review, the paper was limited by its methods as pulling together all the results of different hallucinogens, such as psilocybin, MDMA, DMT or LSD ,and there effects on the depression. The authors should tone down their conclusion ("strongly suggest") on the effects of these chemicals on depression, since most of the numbers of participants of these 14 studies, either randomized control trials or open-labled ones, most of them were less than 100, even lower than 50.
2. However, there is a strength of this paper: it include the analysis of the effects of several "designer drugs", that is, MDMA and DMT. For examples, Romeo et al (2020) and Galvão-Coelho (2021) have published paper by meta-analysis on similar topic, but they mostly analyzed the effects of LSD and psilocybin. Authors are advised to compare your results with those studies on the similar issues:
Romeo B, Karila L, Martelli C, Benyamina A. Efficacy of psychedelic treatments on depressive symptoms: A meta-analysis. J Psychopharmacol. 2020;34(10):1079-1085. doi: 10.1177/0269881120919957.
Galvão-Coelho, N.L., Marx, W., Gonzalez, M. et al. Classic serotonergic psychedelics for mood and depressive symptoms: a meta-analysis of mood disorder patients and healthy participants. Psychopharmacology 238, 341–354 (2021). https://doi.org/10.1007/s00213-020-05719-1
3. Furthermore, in the Introduction, authors might also briefly review the reasons why these chemicals are not inluded into the armamentarium for the treatment of depression. The psychedelic effects ? The abuse potential? The risk of tolerance or dependence? Do they harm more than benefit? This brief review would be helpful for biomedlcal or life scientists who are not familiar with the field of psychological medicine or psychiatry.
4. Last but not least, authors have searched 14 previous trials for this topic. Could they consider conducting a meta-analysis of these trials? An SMD could serve as evidence, which would more helpful to support the potential beneficial effects of hallucinogens on depressive disorders.
Author Response
Dear Reviewer,
Thank you for your suggestions and supplementary comments.
Ad.1.: In the redrafted version of the article, we have tempered the enthusiastic tone that appeared mainly in conclusions. (Discussion, 324-327; Conclusions, 408-410)
Ad.2.: We have also referred to recommended articles, in the second paragraph of Discussion (Discussion, 297-318).
Ad.3.: It is difficult to address all the reasons why psychedelic substances are not available as depression treatment yet. First of all, more extensive, large-scale surveys must be done in this regard, in order to assess and definitely confirm efficacy and safety of these psychedelic substances. As for your suggestions, we have addressed most of them in the discussion (Introduction, 75-77; Discussion, 369-373; 385-391)
Ad.4.: We consider conducting meta-analysis of these trials in future.
Kind Regards
Authors
Reviewer 2 Report
In the intro (1.2), the role of pharmacological treatment should be reported as the first possible treatment, given the higher treatment response showed in clinical practice and metanalysis.
In the paragraph 1.3 the authors, among psychedelics, reported the role of MDMA as a potential antidepressant. Although its potentialities in the area of depressive disorders can be of interest, the authors should acknowledge that until now most of the studies in the literature are reporting detrimental effects of this substance (psychosis, mood alterations, aggressivity, impulsiveness, suicidality, serotonin syndrome, HPPD, fatal hyperthermia, etc..). This position in this paragraph is not acceptable and should be abundantly revised.
What do the authors mean for “cranial electrotherapy stimulation”? brain stimulation? This point should be clarified. However, the role of rtms and tdcs should be better reported, given the efficacy showed in treatment resistant depression.
The fundamental fact that in most of the studies reported the subjects included did not represent severe form of depression, and much less treatment resistant cases should be clearly reported.
In the discussion the enthusiasm showed by the authors when reporting the comparison between antidepressants and psychedelic therapy should be tempered: the sample size of psychedelic studies was small in each study, we only have one comparison trial, in some studies we have the participation of healthy volunteers rather than severe or resistant patients affected by major depression.
Although psychedelic treatments are generally well tolerated, possible risk cannot be ruled out. Hallucinogens are involved in cases of serotonergic syndrome (Schifano F, et al. New psychoactive substances (NPS) and serotonin syndrome onset: A systematic review. Exp Neurol. 2021), HPPD (Litjens RP, et al. Eur Neuropsychopharmacol. 2014), exogenous psychosis (Martinotti G, et al. Substance-related exogenous psychosis: a postmodern syndrome. CNS Spectr. 2021) and other detrimental effects (Docherty JR, Green AR. The role of monoamines in the changes in body temperature induced by 3,4-methylenedioxymethamphetamine (MDMA, ecstasy) and its derivatives. Br J Pharmacol. 2010) that should still be considered, at least before further studies with larger sample will show it in short- and long-term follow-up. These considerations and possible risks must be reported in the discussion session.
Author Response
Dear Reviewer,
Thank you for your suggestions and supplementary comments. We answered them below (Red text).
In the intro (1.2), the role of pharmacological treatment should be reported as the first possible treatment, given the higher treatment response showed in clinical practice and metanalysis.
Response 1: In the redrafted version of the article, we noted the superior importance of using pharmacological treatment and psychotherapy (Discussion, 325-327; Conclusions, 408-409)
In the paragraph 1.3 the authors, among psychedelics, reported the role of MDMA as a potential antidepressant. Although its potentialities in the area of depressive disorders can be of interest, the authors should acknowledge that until now most of the studies in the literature are reporting detrimental effects of this substance (psychosis, mood alterations, aggressivity, impulsiveness, suicidality, serotonin syndrome, HPPD, fatal hyperthermia, etc..). This position in this paragraph is not acceptable and should be abundantly revised.
Response 2: We mentioned the cases of serious adverse events caused by psychedelics in Discussion (340-347). However, we did not manage to find any reliable human studies which report mood impairment, impulsiveness or aggressiveness caused by MDMA administration. In aggressiveness case, we found only studies on mice and rats (Miczek, K. A., & Haney, M. (1994). Psychomotor stimulant effects of d-amphetamine, MDMA and PCP: aggressive and schedule-controlled behavior in mice. Psychopharmacology, 115(3), 358–365. https://doi.org/10.1007/BF02245077; Wallinga, A. E., ten Voorde, A. M., de Boer, S. F., Koolhaas, J. M., & Buwalda, B. (2009). MDMA-induced serotonergic neurotoxicity enhances aggressiveness in low- but not high-aggressive rats. European journal of pharmacology, 618(1-3), 22–27. https://doi.org/10.1016/j.ejphar.2009.07.006; De-Giorgio, F., Bilel, S., Ossato, A., Tirri, M., Arfè, R., Foti, F., Serpelloni, G., Frisoni, P., Neri, M., & Marti, M. (2019). Acute and repeated administration of MDPV increases aggressive behavior in mice: forensic implications. International journal of legal medicine, 133(6), 1797–1808. https://doi.org/10.1007/s00414-019-02092-3).
What do the authors mean for “cranial electrotherapy stimulation”? brain stimulation? This point should be clarified. However, the role of rtms and tdcs should be better reported, given the efficacy showed in treatment resistant depression.
Response 3: Cranial Electrotherapy Stimulation (CES) is brain stimulation, a technique approved for resistant depression treatment. As for your suggestions, we mentioned RTMS as FDA-approved method for treatment resistant MDD (Introduction, 58-62).
The fundamental fact that in most of the studies reported the subjects included did not represent severe form of depression, and much less treatment resistant cases should be clearly reported.
Response 4: Characteristics of each reviewed study, with characteristics of participants, are showed in Table 1. and Table 2.
In the discussion the enthusiasm showed by the authors when reporting the comparison between antidepressants and psychedelic therapy should be tempered: the sample size of psychedelic studies was small in each study, we only have one comparison trial, in some studies we have the participation of healthy volunteers rather than severe or resistant patients affected by major depression.
Response 5: In the redrafted version of the article, we tempered the enthusiastic tone that appeared mainly in conclusions. (Discussion, 324-327; Conclusions, 408-410)
Although psychedelic treatments are generally well tolerated, possible risk cannot be ruled out. Hallucinogens are involved in cases of serotonergic syndrome (Schifano F, et al. New psychoactive substances (NPS) and serotonin syndrome onset: A systematic review. Exp Neurol. 2021), HPPD (Litjens RP, et al. Eur Neuropsychopharmacol. 2014), exogenous psychosis (Martinotti G, et al. Substance-related exogenous psychosis: a postmodern syndrome. CNS Spectr. 2021) and other detrimental effects (Docherty JR, Green AR. The role of monoamines in the changes in body temperature induced by 3,4-methylenedioxymethamphetamine (MDMA, ecstasy) and its derivatives. Br J Pharmacol. 2010) that should still be considered, at least before further studies with larger sample will show it in short- and long-term follow-up. These considerations and possible risks must be reported in the discussion session.
Response 6: In the redrafted version of the article, we included the cases of serious adverse events caused by psychedelics in Discussion (340-347).
Kind regards,
Authors
Round 2
Reviewer 2 Report
The authors addressed all the points raised